# Early Intervention of Cold-Water Swimming on Functional Recovery and Spinal Pain Modulation Following Brachial Plexus Avulsion in Rats

**DOI:** 10.3390/ijms23031178

**Published:** 2022-01-21

**Authors:** Yueh-Ling Hsieh, Nian-Pu Yang, Shih-Fong Chen, Yu-Lin Lu, Chen-Chia Yang

**Affiliations:** 1Department of Physical Therapy, Graduate Institute of Rehabilitation Science, China Medical University, Taichung 406040, Taiwan; ul.family10@gmail.com; 2Kao-An Physical Medicine and Rehabilitation Clinic, Taichung 40763, Taiwan; cmuhsieh@gmail.com (N.-P.Y.); kaoanptc@gmail.com (S.-F.C.); s901100@gmail.com (C.-C.Y.)

**Keywords:** brachial plexus avulsion, cold-water swimming, functional recovery, pain modulation

## Abstract

Brachial plexus avulsion (BPA) causes peripheral nerve injury complications with motor and sensory dysfunction of the upper limb. Growing evidence has shown an active role played by cold-water swimming (CWS) in alleviating peripheral neuropathic pain and functional recovery. This study examined whether CWS could promote functional recovery and pain modulation through the reduction of neuroinflammation and microglial overactivation in dorsal horn neurons at the early-stage of BPA. After BPA surgery was performed on rats, they were assigned to CWS or sham training for 5 min twice a day for two weeks. Functional behavioral responses were tested before and after BPA surgery, and each week during training. Results after the two-week training program showed significant improvements in BPA-induced motor and sensory loss (*p* < 0.05), lower inflammatory cell infiltration, and vacuole formation in injured nerves among the BPA–CWS group. Moreover, BPA significantly increased the expression of SP and IBA1 in dorsal horn neurons (*p* < 0.05), whereas CWS prevented their overexpression in the BPA–CWS group. The present findings evidenced beneficial rehabilitative effects of CWS on functional recovery and pain modulation at early-stage BPA. The beneficial effects are partially related to inflammatory suppression and spinal modulation. The synergistic role of CWS combined with other management approaches merits further investigation.

## 1. Introduction

Brachial plexus avulsion (BPA) is considered the most severe type of injury to the upper limb. It is usually caused by high-energy trauma with a tremendous amount of stretching force and is often combined with multiple injuries of one or more roots from the spinal cord levels of C5 to T1 [1,2]. An avulsion injury triggers intense neuroinflammation at the lesion site both in the peripheral and central nervous systems; therefore, BPA has unique characteristics including motor and sensory dysfunction of the upper limb, and even total loss of these functions during the whole pathophysiological process of global avulsion [3]. In view of its serious impact on the patient’s quality of life, early intervention of BPA is often necessary. 

In clinical practice, it remains difficult to treat BPA. On one hand, there is a lack of knowledge on the underlying degenerative mechanisms of BPA; on the other hand, most current treatments fail to effectively alleviate sensory impairments and pain. Central sensitization of the spinal cord is known to be caused by an increase in inflammatory cytokines of injured peripheral nerves, substance P (SP) immunoexpression in spinal cord astrocytes, microglia and neurons [4]. In particular, early-stage BPA pain is the result of inflammation and microglial activation in the spinal cord [2]. Effective modulation of BPA-induced neuropathic pain by inhibiting spinal glial cells and microglial pathways have been reported [2,5,6].

Prior research has evidenced increased axonal regeneration and improved functional recovery through physical exercise after peripheral nerve injury [7,8,9]. A commonly used exercise training for functional recovery after nerve injury is swimming. In rats, swimming is a naturally occurring behavior with features similar to those of ambulation. In the case of peripheral nerve, spinal cord and brain injuries, hypothermia was found to improve nerve tissue sparing. Its neuroprotective effects are attributed to overall reduction in blood flow, oxygen consumption, metabolic activity, and inflammation [9,10,11,12]. In view of these benefits after BPA, swimming in water at temperatures lower than the animal’s core temperature, cold-water swimming (CWS), may contribute to improvement of functional recovery.

The present study established a BPA model in adult rats with sensory and motor dysfunction confirmed by behavioral tests. The main objective was to examine whether CWS had beneficial effects in promoting functional recovery and modulating the spinal mechanism evaluated by behavioral tests, morphological and immunohistochemistry studies of SP (for pain transmission) and ionized calcium-binding adaptor molecule 1 (IBA1; for microglia) at the early stage of BPA. Our hypothesis proposes that CWS intervention at early-phase BPA can enhance functional recovery and trigger spinal modulation of neuropathic pain. 

## 2. Results

### 2.1. Effects of CWS on Motor Function and Sensory Behavioral Assessments in BPA Rats

Prior to the BPA operation, there were no significant differences in any of the functional and behavioral tests among the four groups (all *p* > 0.05).

#### 2.1.1. Motor Function

Serial alterations of the front limb’s grip strength on motor function for the four groups are shown in Figure 1A. In a grip strength assessment, the sham BPA controls, including both sBPA–CWS (χ^2^(3) = 4.21, *p* = 0.24) and sBPA–sCWS (χ^2^(3) = 7.41, *p* = 0.06) groups, showed no significant change in grip force until the end of the experiment (*p* > 0.05). In contrast, marked changes were observed in both BPA–CWS (χ^2^(3) = 25.29, *p* < 0.0001) and BPA–sCWS (χ^2^(3) = 30.00, *p* < 0.0001) groups. BPA significantly reduced the grip force of the front limb compared with those receiving sham BPA at the time points of Post-op (Z = −5.79, *p* < 0.0001), 1 week post training (1Wposttr) (Z= −5.53, *p* < 0.0001), and 2 weeks post training (2Wposttr) (Z= −5.53, *p* < 0.0001). The loss of grip force induced by BPA improved significantly after repeated application of CWS in the BPA–CWS group compared with the BPA–sCWS group at assessment time points of 1Wposttr (Z= −2.80, *p* = 0.023) and 2Wposttr (Z= −2.80, *p* = 0.023, Figure 1A).

#### 2.1.2. Mechanical Sensitivity

Serial alterations of paw withdrawal thresholds on mechanical sensitivity tested using von Frey filaments in BPA rats after CWS treatments are shown in Figure 1B. As can be seen, the sham BPA controls, including both sBPA–CWS (χ^2^(3) = 6, *p* = 0.11) and sBPA–sCWS (χ^2^(3) = 7.2, *p* = 0.06) groups, showed insignificant reduction in withdrawal latencies until the end of the experiment. In contrast, BPA significantly elevated the threshold of paw withdrawal responses compared with sham BPA at time points of Post-op (Z = −5.78, *p* < 0.0001), 1Wposttr (Z= −4.89, *p* < 0.0001) and 2Wposttr (Z= −4.90, *p* < 0.0001). Such an increase in withdrawal threshold induced by BPA was markedly reduced after repeated application of CWS in the BPA–CWS group compared with the BPA–sCWS group at 2Wposttr (Z= −2.15, *p* = 0.04, Figure 1B).

#### 2.1.3. Cold Sensitivity

Serial alterations of paw withdrawal latencies on cold sensitivity tested by acetone stimulation in BPA rats after CWS treatments are shown in Figure 1C. In the acetone test, the sham BPA controls, including both sBPA–CWS (χ^2^(3) = 3.24, *p* = 0.36) and sBPA–sCWS (χ^2^(3) = 6.12, *p* = 0.11) groups, showed insignificant changes of withdrawal latencies until the end of the experiment (*p* > 0.05). BPA markedly prolonged the onset latency of withdrawal responses in the acetone test compared with sham BPA at time points of Post-op (Z = −5.79, *p* < 0.0001), 1Wposttr (Z= −5.46, *p* < 0.0001) and 2Wposttr (Z= −5.46, *p* < 0.0001). Withdrawal latencies tested by acetone remained persistently prolonged up to 2 weeks in both BPA–CWS and BPA–sCWS groups. CWS significantly reduced the withdrawal latency prolonged by BPA in the BPA–CWS group at time points of 1Wposttr (Z= −2.47, *p* = 0.02) and 2Wposttr (Z= −2.47, *p* = 0.02) compared with the BPA–sCWS group when exposed to acetone stimulus.

#### 2.1.4. Thermal Sensitivity 

Serial alterations of reaction latencies on thermal sensitivity tested by menthol and capsaicin stimulation in BPA rats after CWS treatments are shown in Figure 1D,E, respectively. Menthol (a TRPM8 agonist) and capsaicin (a TRPV1 agonist) provided stimulation to the plantar surface of the hind paw which significantly enhanced the onset latency of withdrawal nocifensive behaviors in the BPA group compared with the sham BPA controls at time points of Post-op (TRPM8: Z = −5.79, *p* < 0.0001; TRPV1: Z = −5.42, *p* < 0.0001), 1Wposttr (TRPM8: Z = −5.42, *p* = 0.94; TRPV1: Z = −5.41, *p* < 0.0001) and 2Wposttr (TRPM8: Z = −5.42, *p* < 0.0001; TRPV1: Z = −5.41, *p* < 0.0001). Significant reduction in BPA-induced prolonged latency of menthol- and capsaicin-evoked withdrawal responses appeared after CWS at time points of 1Wposttr (TRPM8: Z = −3.57, *p* < 0.001; TRPV1: Z = −3.79, *p* < 0.001) and 2Wposttr (TRPM8: Z = −3.79, *p* < 0.001; TRPV1: Z = −3.79, *p* < 0.001) in the BPA–CWS group compared with the BPA–sCWS group.

### 2.2. Effects of CWS on Morphological Studies in Brachial Plexus in BPA Rats

Figure 2A–D displays morphological changes in longitudinal cuts of transected nerves of the four groups, respectively. After H&E staining, both BPA–CWS (Figure 2A) and BPA–sCWS (Figure 2B) groups showed diffusely increased nuclei percentage with more inflamed cells and vacuoles formed after BPA in comparison with both sBPA–CWS (Figure 2C) and sBPA–sCWS (Figure 2D) groups (Z = −5.41, *p* < 0.001). Moreover, infiltration of immune cells and BPA-induced vacuole formation were significantly decreased in the BPA–CWS group compared with the BPA–sCWS group (both Z = −3.79, *p* < 0.001). No marked difference in inflammatory cell infiltration and vacuole formation was observed between the sBPA–CWS and sBPA–sCWS groups (Z = −1.38, *p* = 0.19, Figure 2E,F). 

### 2.3. Effects of CWS on Protein Levels of SP and IBA1 in Superficial Dorsal Horns in BPA Rats

Significant differences in expression of SP-LI (Figure 3A–D) and IBA1-LI (Figure 3F–I) in the spinal dorsal horn were observed among the four groups (SP-LI: χ^2^(3) = 34.14, *p* < 0.001; IBA1-LI: χ^2^(3) = 33.00, *p* < 0.001; Figure 3E, J). Expressions of SP-LI and IBA1-LI in BPA groups (BPA–CWS and BPA–sCWS groups) were significantly increased when compared with those in sham BPA groups (sBPA–CWS and sBPA–sCWS groups) (SP-LI: Z = −5.41, *p* < 0.001; IBA1-LI: Z = −5.28, *p* < 0.001). CWS significantly reduced SP-LI cells and IBA1-LI in dorsal horns, especially in Lamina I and II in the BPA–CWS group compared with the BPA–sCWS group (SP-LI: Z = −3.79, *p* < 0.001, Figure 3E; IBA1-LI: Z = −3.79, *p* < 0.001, Figure 3J).

## 3. Discussion

In this study, significant hypoalgesia and hyposensitivity were observed in animals examined using all sensory behavioral tests after BPA. However, two weeks of daily CWS training in rats at early phase of BPA led to improvements in at least four aspects of functional recovery including grip strength, mechanical, cold, and thermal thresholds compared with untrained rats bearing the same injury. Furthermore, morphological and immunohistochemical measurements demonstrated improvements, including reduction of inflammatory cell infiltration and vacuole formation in injured nerves and decreased expression of SP and IBA1 in spinal dorsal horns corresponding to BPA levels, indicating CWS as a potential therapy for reducing inflammation and pain at early stage of peripheral nerve injury. Many studies reported better functional recovery by cell transplantation, immunotherapy, or nerve transfers when other forms of therapeutic interventions were also applied for management of severe peripheral nerve injury [13,14,15,16]. The present findings evidenced the potential of CWS in enhancing sensory and motor functional recovery, in addition to modulating BPA-induced neuropathic pain. Moreover, it may be used in combination with other pharmacological therapies in treating neuropathic pain (e.g., nerve block anesthesia) or nerve reconstruction (e.g., cell-based therapy) for early functional mobilization [9,16,17,18] to achieve greater beneficial restoration than a single treatment approach for management of BPA. 

For promoting functional recovery after peripheral nerve injury, diverse therapeutic interventions through physical exercise have been studied [17,19,20]. The two most common exercises used are treadmill running and swimming. Both were found to enhance physical activity through fostering neurogenesis and neurotrophic expression, reducing apoptosis and inflammation as well as improving neurovascular integrity. By activating these different processes, physical exercise training has neuroprotective effects in both central and peripheral nervous systems [20]. However, strenuous exercise was found to increase pain sensation and impede functional recovery in animals with peripheral nerve injury [21,22]. Hence, the intensity of training affects the outcome of exercise intervention. A previous study reported behavioral and physiological changes as well as inhibited functional recovery in rats put on treadmill fitted with electric foot shocks, which induced a stress response [23]. In comparison, animals experience less stressful stimulation when swimming. Rats showed an adaptive response to stress when placed in water. After swimming energetically for several minutes, they would eventually stop and remain afloat by moving just enough to hold their heads above the water surface [24].

Swimming presents favorable results in functional recovery, axonal diameter recovery [9,25], stimulation of neuronal growth and acceleration of nerve regeneration [26]. However, conflicting effects of swimming on nerve regeneration were reported. While swimming exercise after nerve crush injury accelerated nerve regeneration with an increase in diameter of nerve fibers [27], daily 2-h intense swimming therapy failed to enhance restoration of muscle innervation of a crushed sciatic nerve [7,28]. Moreover, animals with sciatic nerve injury submitted to the swimming protocol (30 min, 45 sessions) after the nerve grafting procedure did not present differences in value of the sciatic functional index and number of motor neurons when compared with the non-swimming (control) group [29]. Moreover, our previous study demonstrated that long-duration swimming induced an exaggerated stress response and impeded functional recovery after sciatic nerve transection lesions were treated with mesenchymal stem cells [21]. Thus, duration and exercise type are more important factors in a recovery swimming exercise model. In this study, the recovery of sensory function was more prominent in the BPA-affected limb than in control limbs, but the recovery of motor function was limited after very short forced swimming training. Short forced swimming exercises seem to provide a less stressful effect on nerve regeneration; thus, swimming was the preferred exercise type in this study. 

BPA-induced neuropathic pain is largely characterized by a quick-onset and long-lasting pain, which is often intolerable and has no known effective treatments [18]. Moreover, it further deteriorates quality of life in patients impaired with motor, sensory and autonomic deficits [3]. Exercise may provide a promising non-pharmacologic therapy for preventing the development of neuropathic pain [30]. In animal models, exercise was found to provoke increased mechanical withdrawal threshold and thermal withdrawal latency. However, previous research has shown that long-duration treadmill running induced worsening of allodynia, while intense short-lasting exercise reduced mechanical allodynia in the neuropathic pain model [22]. Swim stress-induced analgesia (SSIA) is a frequently studied form of pain modulation assessed in rats and mice upon completion of forced swimming, especially in swimming under a hypothermic environment [31,32]. A previous study found that swimming training decreased peripheral neuropathic pain marked by increased thermal withdrawal latencies and mechanical von Frey thresholds, as well as reduced levels of proinflammatory cytokines (TNF-α and IL-1β) after chronic constriction injury of the sciatic nerve [33]. The present results demonstrate that CWS-induced reduction of inflammatory cell infiltration in injured nerves subsequently suppresses expression of SP aggregation in the dorsal horn of spinal cords, suggesting that CWS has beneficial anti-inflammatory effects and contributes to neuropathic pain resolution in BPA animals. 

Neuropathic pain following peripheral nerve injury is caused by the development of spinal microgliosis triggered by nociceptive inputs from sensory neurons [34]. A higher density of microglia were observed not only within the dorsal horn of the spinal cord at the location of injured sensory afferent terminals but also within the ventral horn around the cell bodies of injured motor neurons [35]. IBA1 is a microglia/macrophage-specific calcium-binding protein with actin-bundling activity and participates in membrane ruffling and phagocytosis in activated microglia [36]. In the BPA animal model [2], increased expression of IBA1 was found on day 1 post-injury and persisted for 28 days. A behavioral test of mechanical stimulation showed that neuropathic pain developed as a result of significant decreases in pain thresholds of bilateral hind limbs. In other words, the BPA model can simulate the development of persistent neuropathic pain [2]. The present results revealed that BPA induced an overexpression of SP and IBA1 in the dorsal horns of spinal cords corresponding to the severity of the injured nerve. CWS training significantly reduced SP expression and ameliorated the BPA-induced overactivation of microglia, thus modulating IBA1 expression in dorsal horns. These biochemical results suggested that CWS reduced the development of neuropathic pain after BPA by modulating pain transmission and microgliosis. Moreover, several studies demonstrated that rats were acutely stressed which might present discrepancies between behavioral manifestation of pain and biochemical induction in the brain and spinal cords. That is, in the presence of stress, the analgesia observed by biochemical expression as indicative of neuroplasticity changes in the pain pathway may not be reflected by hyperalgesic sensory behavioral assessments [37,38,39]. Our results were in agreement with those of previous studies, which demonstrated how CWS ameliorated acutely stressful experiences caused by altered responsivity in dorsal horns of the spinal cord, as evidenced by decreased SP and IBA1 expression although hypoalgesic behaviors in BPA animals were not expressed. 

Hypothermia was shown to suppress inflammation, inhibit the extrinsic cell death pathway and reduce abnormal receptor activity through a hypoxic–ischemic neuroprotective strategy [40]. Mounting studies have demonstrated that rats manifest SSIA only if the swimming conditions are combined with hypothermia [31,41,42]. Therefore, to induce hypothermia, the experimental animals were subjected to CWS, i.e., swimming at a water temperature below the animal’s core temperature. A markedly lower magnitude of SSIA was observed in mice accustomed to swimming in 20°C than in 32°C water, suggesting that an interaction between the emergency and hypothermic components of swim stress resulted in SSIA [43]. Evidence shows that repeated immersion in ice water (acute exposure to CWS) produces analgesia, but chronic exposure over daily sessions leaves nociceptive thresholds unaltered [44], suggesting that SSIA adapts in much the same manner as autonomic and neuroendocrine stress responses following chronic exposure to stressors [45]. Significant core and skin temperature reductions occur following both acute and chronic exposure to CWS, however, only acute exposure results in analgesia [45]. In the present study, animals acutely exposed to CWS for 5 min only triggered an analgesic response, thus reducing SP overexpression in cervical dorsal horns. In addition, our previous study demonstrated that swimming exercise training at room temperature had no synergistic effect on functional recovery in transected nerves treated with mesenchymal stem cell transplantation [21]. Nevertheless, improved functional recovery in nerve crush injury could be achieved by combining CWS with stem cell transplantation than by treatment with mesenchymal stem cell or CWS alone [9]. This study found a more severe inflammatory cell invasion in BPA-operated rats than in sham-operated rats, and significant reduction in inflammatory cells in BPA-operated rats treated with CWS than in those subjected to sham CWS. Taken together, these results evidenced suppression of BPA-induced inflammatory cell accumulation through CWS intervention, which resulted in enhanced functional recovery. Our findings are consistent with previous results of animal studies in that CWS training attenuated the reaction latency of heat, cold and mechanical stimuli in rats after BPA and suppressed overexpression of SP and IBA1 in dorsal horns of the cervical spinal cord.

To increase fatigue resistance and restore the contractile properties and mechanical sensitivity of muscles, physical activity should resume immediately after nerve injury or at early-stage denervation [46]. Moreover, physical activities help prevent neurotrophin-mediated hyperexcitability of injured and uninjured sensory neurons, which causes neuropathic pain [47]. When applied during the acute or late phase of nerve injury, swimming was reported to accelerate nerve regeneration and synaptic elimination after axonotmesis [27]. Moreover, it promoted early initiation of regrowth, self-fusion of proximal and distal ends, as well as post-regrowth enhancement of function [48]. Furthermore, ischemia-reperfusion peripheral nerve injury may worsen with prolonged delay of hypothermic treatment [49]. Echoing these findings, the present results confirm the benefits of early intervention of hypothermia and swimming for functional recovery in rats with BPA. 

Some limitations should be taken into account. First, the brief period of very moderate CWS exercise used in the present study, while sufficient to promote sensory recovery and modest functional recovery, remains suboptimal to enhancing full functional recovery, particularly regarding grip strength. Although not resulting in a return to full functional recovery, at least as indicated by sensory behavioral test results, the enhancements of sensory function undoubtedly contribute to a restored ability of the animals to respond to different demands of stimuli sensation. Different paradigms including progressive intensity of swimming protocols exhibiting a more pronounced effect on motor functional recovery merit further study. Second, this biochemical study assessed the underlying pain modulation mechanism of CWS in BPA-induced neuropathy only within spinal cord levels. According to previous studies, the action of CWS-induced stress may also involve mechanisms of opioid and non-opioid SSIA [31,41]. More advanced studies will be required to reveal supraspinal pain modulation involvement that can provide more information of such neuroplasticity alterations of opioid and non-opioid systems with CWS applied to management of BPA. Finally, this study lacked a long-term assessment. Aspects related to fluctuation periods and pain remission in patients with BPA must be considered before any final conclusions can be made on therapeutic efficacy which warrants further investigation. 

## 4. Materials and Methods

### 4.1. Study Design

Figure 4 summarizes the study design. Based upon an a priori power sample size calculation, a total of 40 rats were used with a power of 0.95 with effect size of 0.69 and an alpha of 0.05 for assessment (analysis of covariance, G*Power, v. 3.1.9.7). There were four experimental groups, each with 10 rats. These rats were randomly assigned to receive BPA or sham BPA (sBPA) and treated with CWS or sham CWS (sCWS). Simple randomization using a computer-generated random number table (Excel RAND function), allocation concealment, and blinded outcome assessment were used to reduce the risk of bias. 

First, BPA surgery was performed on a randomly selected unilateral brachial plexus. Three days after surgery, two sessions of 5-min swimming training daily commenced, lasting for two weeks. At four time points, namely before (Pre-op), immediately after surgery (Post-op), at first and second weeks after treatments (1Wposttr and 2Wposttr), the rats underwent motor function and sensory behavioral tests to assess functional recovery. Upon completion of CWS treatment, the rats were sacrificed for morphological and immunohistochemistry studies. 

### 4.2. Animal Care and Preparation 

Adult male Sprague–Dawley rats (SD, 250 to 300 g) were purchased from BioLASCO Co., Ltd., Taiwan, China. Prior to the experiment, rats were housed individually in standard animal rooms at 22 °C with free access to food pellets and water. In compliance with the ethical guidelines of the International Association for Study of Pain in animals [50], they were cared for with minimum discomfort by the same animal breeder to ensure consistent feeding and management for all groups. In accordance with the Guidelines for Animal Experimentation (CMUIACUC-2019-089), the experiments were conducted following procedures approved by the Animal Care and Use Committee of China Medical University. 

### 4.3. Surgical Procedures for Brachial Plexus Injury

The surgical procedures used in this study followed that described by Rodrigues-Filho et al., [51]. Animals were anesthetized with 7% chloral hydrate (0.6 mL/kg; i.p.). Then a horizontal 1-cm incision was made parallel to the clavicle, running from the sternum to the axillary region. The right brachial plexus was approached with the subclavian vessels located and the lower trunk exposed. For the BPA groups, the lower trunk was grasped with forceps and extracted from the spinal cord by traction; for the sBPA groups, the brachial plexus was exposed with no lesions made to the nerve. The surgery was finished with tissue layers brought together and the skin closed with a 4-0 silk suture string (Ethicon, Edinburgh, Scotland).

### 4.4. Swimming Intervention

The swimming intervention protocol followed that in our earlier study (Wang et al., 2010). The swimming exercise was conducted in a plastic container with dimensions of 60 cm (H) × 35 cm (W) × 60 cm (L). For each swimming session, it was filled with icy water (10 °C lower than room temperature, average 19.5°) to an approximate depth of 40 cm. A drop of liquid soap was added to reduce surface tension, which additionally decreased the frequency of “floating” behavior, thus ensuring a full swimming session with right/left alternation of flexion and extension by front limb. The container was thoroughly cleaned daily. 

For habituation, each rat underwent two sessions of three- to five-minute swimming daily for one week. Three days after surgery, swimming intervention commenced. For the CWS groups, each rat underwent two sessions of five-minute swimming daily, in the morning (7:00) and afternoon (17:00), for two weeks. 

After each swimming session, the rats were gently dried with a cloth towel and their limbs were left free for slow rewarming at room temperature. Deep rectal temperature was measured with a thermistor probe and maintained at 36.0 ± 1.0 °C using an infrared lamp with the temperature clamped by servo control. 

### 4.5. Functional Assessments

Prior to testing, each rat was placed inside an acrylic cage with dimensions of 32 cm (H) × 22 cm (W) × 27 cm (L) and a wire grid floor in a quiet room for 30-min habituation. All functional and behavioral assessments were performed at room temperature (27 °C) between 09:00 and 16:00. The same experimenter who was blinded to the treatment condition conducted all the tests. 

#### 4.5.1. Motor Function Test

Motor strength was tested using the grip strength test meter (IITC Life Science Inc., Woodland Hills, CA, USA), which comprises a force transducer that is connected to a wire mesh grid connected to an anodized base plate. In this test, the rat was held at the lower trunk and hind paws and was gently passed over the mesh until it grasped the grid with its front paws. Muscle strength was defined as the peak pull-force (*g*). Five grip force measurements were taken and then averaged to determine the grip force of front limbs.

#### 4.5.2. Sensory Behavioral Examinations

##### Mechanical Sensitivity

To assess mechanical sensitivity, paw withdrawal threshold in response to pressure stimulation with an electronic von Frey anesthesiometer (IITC Life Science) was measured as previously described [52]. Initial pressure of 20 log of force (*g*) was applied on the filament, followed by a gradual increase in pressure until the rat withdrew its front paw. The lowest pressure eliciting the withdrawal reflex response denotes the mechanical paw withdrawal threshold (*g*). Three measurements were made and then averaged.

##### Cold Sensitivity 

To assess cold sensitivity, the acetone stimulation test which measures the onset latency of withdrawal response after spraying 50 microliters of acetone (Honeywell Burdick & Jackson, Muskegon, MI, USA) onto the plantar skin of each hind paw was performed. Three measurements at 5-min intervals were made and then averaged.

##### Thermal Sensitivity 

To assess thermal sensitivity, menthol (250 μM, #W266523, Sigma-Aldrich, St. Louis, MO, USA) and capsaicin (1 μM, #12084, Sigma-Aldrich, St. Louis, MO, USA) were topically applied onto the plantar surface of the front paw. Menthol, a transient receptor potential melastatin subtype 8 (TRPM8) agonist, evokes a cold sensation; and capsaicin, a transient receptor potential vanilloid subtype 1 (TRPV1) agonist, evokes a hot sensation [53,54]. These sensations induce nocifensive responses manifested as licking, flinching, lifting, and backward walking of the stimulated front paw. The reaction latencies of these responses were measured. Three measurements were made at each of the two nociceptive tests performed at 10-h intervals and then averaged. 

### 4.6. Morphology and Immunoassays

#### 4.6.1. Tissue Preparation and Morphological Examination

All animals from each group were euthanized by anesthetic overdose one day following the final swimming session. The brachial plexus, marked by the C5-T1 levels of spinal cords were harvested and fixed in 4% paraformaldehyde for 2 h at 4 °C and then embedded in paraffin. For light microscopy analysis, 5-μm-thick spinal cord and nerve sections in transverse and sagittal planes cut using a microtome were stained with hematoxylin–eosin and examined under a light microscope to identify the morphology of injured nerves. The area of the inflamed cell and nerve nuclei (%) and vacuole (%) in the H&E-stained sections were measured.

#### 4.6.2. Immunohistochemical Staining and Quantitative Analyses

Spinal cord specimens of 5 μm thickness were serially cut in the sagittal plane using a microtome. Approximately 20 consecutive sections were obtained in each region. First, the location where C5-T1 spinal cords could be recognized was determined, followed by random selection of starting position for sampling. Immunohistochemical staining was performed in five alternate sections selected at a periodic sampling interval of four sections for assay. The sections were incubated overnight at 4 °C first with antibody against SP (1:2000, #20064, ImmunoStar, WI, USA) and IBA1 (1:200, # PA5-27436, Thermo Fisher Scientific, Waltham, MA, USA), then with biotinylated goat anti-rabbit IgG secondary antibody (Jackson ImmunoResearch Laboratories, Inc., West Grove, PA, USA), and finally with a streptavidin-horseradish peroxidase conjugate (Jackson ImmunoResearch Laboratories, Inc., West Grove, PA, USA). After adding 3, 3′-diaminobenzidine (DAB, Pierce, Rockford, IL, USA) as a substrate, brown precipitates were visualized and examined under a light microscope (BX43, Olympus America Inc., NY, USA). Five randomly selected fields in dorsal horn regions of C5-T1 spinal cords selected by an investigator blinded to the experimental condition were photographed using a digital color camera (Auto exposure mode, DP70, Olympus America Inc.). The digital images were analyzed by computer-based morphometry using the ImageScope software package with the Color Deconvolution v9 tool (v9.1.19.1571, Aperio, Vista, CA, USA).

### 4.7. Statistical Analysis

All data are expressed as mean ± standard deviation (SD). The Kolmogorov–Smirnoff test indicated a non-normal distribution of all data in all measurements. The effects of CWS treatment on functional assessments, quantitative analyses of morphological and immunohistochemical studies in the contents of inflammatory cells of injured nerves, and SP and IBA1 in dorsal horns of spinal cords were, respectively examined using a Kruskal–Wallis test to determine the significant differences among the four groups (BPA–CWS, BPA–sCWS, sBPA–CWS and sBPA–sCWS). Post hoc comparisons between two groups were analyzed using the Mann–Whitney test. A Friedman test was performed to determine the differences among the four time points (Pre-op, Post-op, 1Wposttr and 2Wposttr) in each group and a post-hoc analysis was conducted using Wilcoxon signed-rank test. A *p* value of < 0.05 was considered statistically significant. All data were analyzed using SPSS v. 22.0 for Windows (SPSS Inc., Chicago, IL, USA).

## 5. Conclusions

BPA carries serious ramifications from the perspective of permanent paralysis of the extremity, prolonged recuperation, and significant socio-economic impacts. This research found improved sensory and motor functions and reduced neuroinflammation, spinal microglial overactivation, and pain transmission after early intervention of short forced CWS in the BPA animal model, indicating that BPA-induced dysfunction can be modulated by CWS. It is the first study demonstrating the underlying mechanism of early CWS training, which can be a beneficial synergistic agent in the management for BPA. Identification of the critical features of CWS that alleviate neuropathic pain is of great importance and necessity if our findings are to be applied in clinical settings. More biochemical studies may help further reveal a central nociceptive transmission mechanism that can provide additional information of such neuroplasticity alterations attributed to effects of CWS on neuropathic pain induced by BPA.

## Figures and Tables

**Figure 1 ijms-23-01178-f001:**
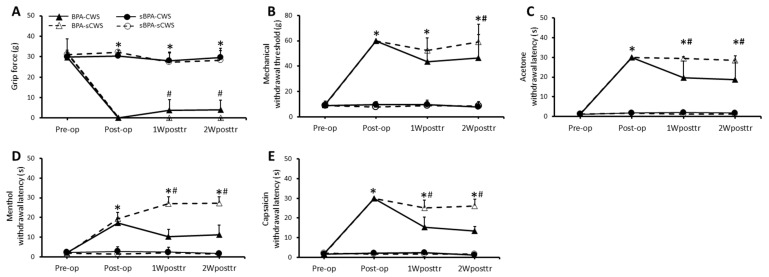
Effects of cold-water swimming (CWS) treatment on grip force of front limbs (**A**), mechanical withdrawal threshold (**B**), cold withdrawal latency (**C**), reaction latencies of menthol—(**D**) and capsaicin—(**E**) evoked nocifensive behavior in the four groups. There were no significant differences among the four groups before surgery; * *p* < 0.05 indicates significant differences among the four groups analyzed using Kruskal–Wallis test. # *p* < 0.05 indicates significant difference between BPA–CWS and BPA–sCWS groups analyzed using Mann–Whitney test.

**Figure 2 ijms-23-01178-f002:**
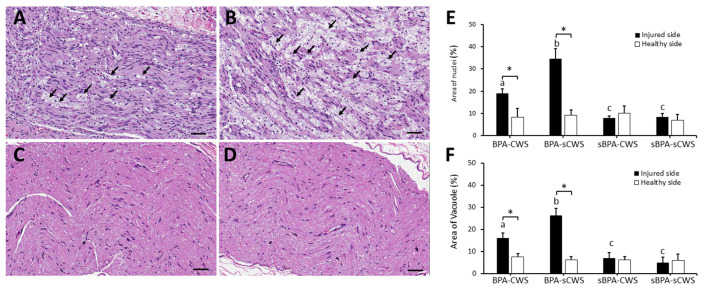
H&E staining in longitudinal cuts of transected nerves of representative animals. The arrows indicate myelin-digestion chambers (vacuole formation). Enhanced vacuole formation and loss of myelinated fibers were found in representative animals treated with BPA–CSW (**A**) and BPA–sCWS (**B**) while intact myelinated fibers were found in representative animals treated with sBPA–CWS (**C**) and sBPA–sCWS (**D**). Data of area of inflammatory cells (**E**) and vacuole formation (**F**) in percentage are presented as mean ± SD. Values with different superscripts (e.g., a vs. b and b vs. c) indicate significant differences (*p* < 0.05) for all possible pairwise comparisons of means in injured sides of all groups analyzed using Mann–Whitney test. * Indicates significant difference (*p* < 0.05) between injured and healthy sides analyzed using paired t-tests. Scale bars, 50 μm.

**Figure 3 ijms-23-01178-f003:**
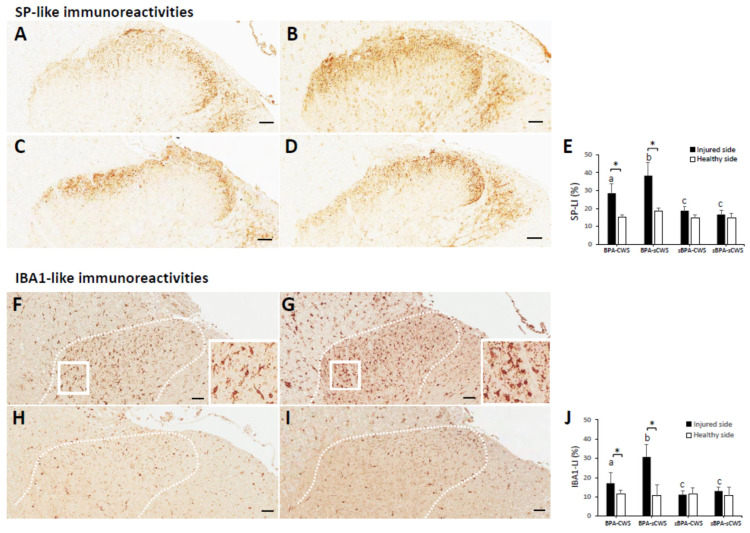
Representative SP-like and IBA1-like immunoreactivities (SP-LI and IBA1-LI) in sections of dorsal horns of C5-T1 in rats of BPA–CWS (**A**,**F**), BPA–sCWS (**B**,**G**), sBPA–CWS (**C**,**H**) and sBPA–sCWS (**D**,**I**) groups. The distributions of SP-LI staining area were mostly located in Laminae I and II while IBA1 expression was located in dorsal horns. Data of SP-LI (**E**) and IBA1 (**J**) in dorsal horns are presented as mean ± SD. Values with different superscripts (e.g., a vs. b and b vs. c) indicate significant differences (*p* < 0.05) for all possible pairwise comparisons of means analyzed using Mann–Whitney tests. * Indicates significant difference (*p* < 0.05) between injured and healthy sides analyzed using paired t-tests. Scale bars, 50 μm.

**Figure 4 ijms-23-01178-f004:**
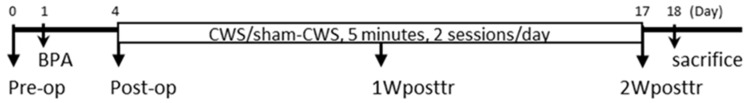
Overview of study design. Rats received brachial plexus avulsion surgery (Day 1) and then were treated with cold-water swimming (CWS) or sham CWS at 3 days after BPA surgery (Day 4) for 2 weeks (ended at Day 17). Assessments of functional recovery included both motor function test and sensory behavioral test performed before (Pre-op, Day 0), immediately after surgery (Post-op, Day 4), at first and second weeks after treatments (1Wposttr and 2Wposttr, Days 10 and 17). Animals were sacrificed for immunoassays (immunohistochemistry and morphological studies) at Day 18.

## Data Availability

Not applicable.

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
