# Peer review of "Early Intervention of Cold-Water Swimming on Functional Recovery and Spinal Pain Modulation Following Brachial Plexus Avulsion in Rats"

_ijms, 2022, doi:10.3390/ijms23031178_

Round 1
Reviewer 1 Report
This study aimed to examine whether cold-water swimming could induce greater functional recovery and modulate spinal mechanism at the early stage of brachial plexus avulsion.
The study has been well designed using a rigorous methodology. The manuscript is well-written, using a very good quality of academic and scientific writing style. I provide few comments to implement the quality of manuscript.
I congratulate for the high quality of research conducted and the evidence provided.
Introduction
Use past tense for objective
Methods
“A sample size of 40 subjects provided a power of 0.95 with effect size of 0.69 and an alpha of 0.05 to detect differences in assessment value. “
Is this a priori sample size calculation? If yes, statistical test should be reported. Please clarify and implement sample size calculation.
Discussion
“pharmacological therapies or neurological surgeries to achieve greater beneficial restoration than a single treatment for management of BPA”
Could authors be more specific about which pharmacological therapies or neurological surgeries could be combined with CWS? Which are the potentially combined mechanisms? Is there already evidence for the pharmacological therapies or neurological surgeries to be cited? Please, implement this part.
Author Response
Reviewer 1
Comment 1: Introduction: Use past tense for objective
Response: the objective section has been revised by using past tense, as “The main objective was to examine whether CWS has beneficial effects in promoting functional recovery and modulating the spinal mechanism evaluated by behavioral tests, morphological and immunohistochemistry studies of SP (for pain transmission) and ionized calcium-binding adaptor molecule 1 (IBA1; for microglia) at the early stage of BPA. Our hypothesis was that CWS intervention at early-phase BPA can enhance functional recovery and trigger spinal modulation on neuropathic pain.”. Please refer to page 2.
Comment 2: Methods: “A sample size of 40 subjects provided a power of 0.95 with effect size of 0.69 and an alpha of 0.05 to detect differences in assessment value. “Is this a priori sample size calculation? If yes, statistical test should be reported. Please clarify and implement sample size calculation.
Response: Yes, the power and effect size calculation is a priori sample size calculation. The detail statistical test has been reported in the sentence as “Based upon an a priori power sample size calculation, a total of 40 rats were used with a power of 0.95 with effect size of 0.69 and an alpha of 0.05 for assessment (analysis of covariance, G*Power, version 3.1.9.7).”. Please refer to page 11. Thank you for your suggestion.
Comment 3: Discussion: “pharmacological therapies or neurological surgeries to achieve greater beneficial restoration than a single treatment for management of BPA”. Could authors be more specific about which pharmacological therapies or neurological surgeries could be combined with CWS? Which are the potentially combined mechanisms? Is there already evidence for the pharmacological therapies or neurological surgeries to be cited? Please, implement this part.
Response: Thank you for your valuable suggestion. The statement has been revised to “It may also be used in combination with other pharmacological therapies in neuropathic pain (e.g., nerve block anesthesia) or nerve reconstruction (e.g., cell-based therapy) for early functional mobilization [9, 16, 18, 29] to achieve greater beneficial resto-ration than a single treatment for management of BPA.”according to reviewer’s suggestion. Please refer to pages 5-6.
Thank you for reviewing our manuscript.

Reviewer 2 Report
The Authors explored the role of cold water swimming in the promotion of functional recovery and pain modulation via reduction of neuroinflammation and microglial overactivation in dorsal horn neurons at early-stage brachial plexus avulsion in rats after surgery. The topic is meaningful and the manuscript is of interest. Minor comment hereby:
M&M
- Please provide more details about the randomization process. In particular, decribe the methods by which the risk of bias was decreased. Allocation concealment? Masking? Other?
Author Response
Reviewer 2
Comment 1. Please provide more details about the randomization process. In particular, describe the methods by which the risk of bias was decreased. Allocation concealment? Masking? Other?
Response: The statement has been added into the Method section for explaining the detail of randomization process and how to reduce the risk of bias, as “Simple randomization by using a computer-generated random number table (Excel RAND function), allocation concealment, and blinded outcome assessment were used to reduce the risk of bias.”. Please refer to page 8. Thank you for your suggestion.
Thank you for reviewing our manuscript.
